# T. cruzi DNA polymerase beta (Tcpolβ) is phosphorylated in vitro by CK1, CK2 and TcAUK1 leading to the potentiation of its DNA synthesis activity

**Edio Maldonado[1]⊚¤\***, **Diego A. Rojas**[2]⊚, **Fabiola Urbina**[1], **Aldo Solari**[1]¤\*

**1** Programa de Biología Celular y Molecular, ICBM, Facultad de Medicina, Universidad de Chile, Santiago, Chile, **2** Instituto de Ciencias Biomédicas (ICB), Facultad de Ciencias de la Salud, Universidad Autónoma de Chile, Santiago, Chile

⊚ These authors contributed equally to this work.
¤ Current address: Programa de Biología Celular y Molecular, Instituto de Ciencias Biomédicas, Facultad de Medicina, Universidad de Chile, Santiago, Chile
\* emaldona@med.uchile.cl (EM); asolari@uchile.cl (AS)

**Data Availability Statement:** All data are in the manuscript and its supporting information files.

**Funding:** This work was funded by grant #1190392 of Fondo Nacional de Desarrollo Científico y

## Abstract

The unicellular protozoan *Trypanosoma cruzi* is the causing agent of Chagas disease which affects several millions of people around the world. The components of the cell signaling pathways in this parasite have not been well studied yet, although its genome can encode several components able to transduce the signals, such as protein kinases and phosphatases. In a previous work we have found that DNA polymerase β (Tcpolβ) can be phosphorylated *in vivo* and this modification activates the synthesis activity of the enzyme. Tcpolβ is kinetoplast-located and is a key enzyme in the DNA base excision repair (BER) system. The polypeptide possesses several consensus phosphorylation sites for several protein kinases, however, a direct phosphorylation of those sites by specific kinases has not been reported yet. Tcpolβ has consensus phosphorylation sites for casein kinase 1 (CK1), casein kinase 2 (CK2) and aurora kinase (AUK). Genes encoding orthologues of those kinases exist in *T. cruzi* and we were able to identify the genes and to express them to investigate whether or no Tcpolβ could be a substrate for *in vitro* phosphorylation by those kinases. Both CK1 and TcAUK1 have auto-phosphorylation activities and they are able to phosphorylate Tcpolβ. CK2 cannot perform auto-phosphorylation of its subunits, however, it was able to phosphorylate Tcpolβ. Pharmacological inhibitors used to inhibit the homologous mammalian kinases can also inhibit the activity of *T. cruzi* kinases, although, at higher concentrations. The phosphorylation events carried out by those kinases can potentiate the DNA polymerase activity of Tcpolβ and it is discussed the role of the phosphorylation on the DNA polymerase and lyase activities of Tcpolβ. Taken altogether, indicates that CK1, CK2 and TcAUK1 can play an *in vivo* role regulating the function of Tcpolβ.

Tecnológico (FONDECYT) (AS) and ICBM Grant (EM). These fundings were used to get all supplies necessary to develop this investigation. The funders had no role in study design, data collection and analysis, decision to publish, or preparation of the manuscript.

## Author summary

*Trypanosoma cruzi* DNA polymerase β (Tcpolβ) can be phosphorylated *in vivo* and this modification potentiates the DNA synthesis activity of the enzyme, which is involved in the DNA base excision repair (BER) system and kDNA replication. However, the protein kinases involved in this process have not been identified yet. In this work, we identified three protein kinases involved in Tcpolβ *in vitro* phosphorylation: CK1, CK2 and TcAUK1. The protein kinase activity of each enzyme was inhibited using specific pharmacological inhibitors. The phosphorylation event on Tcpolβ by the identified protein kinases increases its polymerizing DNA activity and this modification might be important for in vivo TcPolβ function.

## Introduction

Chagas disease is a Neglected Tropical Disease, which affects several million people around the world. It is caused by the unicellular flagellate protozoan *Trypanosoma cruzi* and affects over 8 million people worldwide, causing approximately 50.000 deaths each year [1,2]. Another 70–100 million people living in endemic areas are at risk of infection. For most of the Latin American countries is one of the main public health problems [1,2]. Chagas disease is transmitted by blood-sucking bugs of the subfamily *Triatominiae*. The disease has two successive phases: first is an acute one characterized by high parasitemia and a second chronic phase usually with cardiomyopathy. Most of the infected individuals (60–70%) never develop symptoms or signs associated to the disease, however, the rest of the patients (30–40%) will progress to the chronic phase with a severe cardiomyopathy [3].

Despite all research done on Chagas disease and on the biology of *T. cruzi*, an effective treatment for Chagas disease has not been obtained [4,5]. Neither an effective treatment nor a cure for the severe cardiomyopathy at the chronic phase has been found [4,5]. Chagas disease is treated with both parasite-specific drugs and with symptomatic treatments to the patients. Two drugs have been used for decades to treat Chagas disease, which are benznidazole and nifurtimox [6]. Those drugs are antiparasitic, however, they could inflict various adverse effects to the patients.

A better understanding of the key biological processes of the causing agent would discover new targets to develop novel drugs to treat efficiently Chagas disease [7] One of those processes is the cell signal transduction pathway used by the parasite to differentiate, to proliferate and to adapt to the different environments [8]. Cell signals are usually transduced by protein kinases, which can phosphorylate key proteins involved in those processes [9,10]. Reversible phosphorylation of key functional proteins is one of the most important biological mechanisms to quickly regulate adaptive responses to intra and extracellular signals in several organisms; therefore, protein kinases and phosphatases are involved in those processes. In higher eukaryotes, cell signals are mainly transduced by protein kinases, which can phosphorylate and activate transcriptional activators, which in turn bind to gene promoters of target genes to activate gene expression. However, trypanosomatids cannot regulate gene expression at the transcriptional level and regulation of proteins function should be done at the mRNA processing, translational and post-translational levels, therefore, reversible phosphorylation of key proteins must be one of those mechanisms to regulate adaptive responses to the signals.

Cell signaling pathways of trypanosomatids are different from that of mammals, including structurally different components [8,11,12]. Such characteristics make those components involved a good candidates as therapeutic targets to develop new drugs to treat Chagas disease.

Several cellular signaling pathways have been described in *T. cruzi*, including lipid, calcium, and cyclic AMP signaling [8,11,12].

Since the genome of *T. cruzi* is available many cellular signaling pathways can be evaluated for the presence or absence of components of those pathways and genes can be easily cloned and expressed to study the biochemical properties of the encoded polypeptides [13]. Also, genetic, transcriptomic, proteomic and bioinformatic approaches can be applied to the study of the parasite [14].

The gene encoding DNA polymerase β from *T. cruzi* (Tcpolβ) has been cloned and the recombinant protein was expressed and studied [15–17]. Tcpolβ locates to the kinetoplast of the cell and it can be crosslinked to the DNA of that organelle, however, it cannot be cross-linked to nuclear DNA [18,19]. In replicative forms of the parasite (epimastigotes and amasti-gotes), the enzyme locates at the antipodal sites of the kinetoplast, however, when the parasite is under oxidative stress by hydrogen peroxide, Tcpolβ can be detected in a third focus outside of the kDNA, most likely located at the kinetoflagellar zone [18]. Tcpolβ overexpression into the parasite confers resistance against high doses of hydrogen peroxide, indicating that the enzyme is involved in the process of kDNA repair of DNA damage caused by oxidative lesions. [18]. Tcpolβ can extend a primer on a DNA template, however, it is unable to perform mis-match extension or DNA synthesis through 8-oxoG lesions. It possesses intrinsic 5 -deoxyri-bose phosphate (dRP) lyase activity, which in part contributes to the kDNA repair of oxidative lesions through the BER repair system [16,18]. Tcpolβ can repair 1–6 bases of gaps of a double strand damaged DNA in the presence of additional *T. cruzi* proteins in *in vitro* assays, which could indicate that *in vivo* is part of a multiprotein complex [17]. Also, when parasites are exposed to high doses of hydrogen peroxide, respond overexpressing Tcpolβ. Interestingly, this enzyme exists in two forms, one that is phosphorylated and other which is not. The phos-phorylated form is active in DNA synthesis, whereas the unphosphorylated form is almost inactive [19].

Tcpolβ has multiple predicted phosphorylation sites for several protein kinases, however, it is unknown whether any of those can phosphorylate the enzyme and whether or not the event has any consequence on the activity of the enzyme [17,19]. In an earlier work [17], predicted phosphorylation sites for CK1, CK2, PKA and PKC were found in Tcpolβ, however, those sites are not in similar positions as those found in the mammalian homologue. Using bioinformatic tools, set at high stringency, we were able to detect phosphorylation sites for casein kinase 1 and 2 (CK1, CK2) and aurora kinase (AUK). Using a gapped BLASTP tool we were able to find orthologues for those three protein kinases encoded in the *T. cruzi* genome. We were able to express the genes encoding those enzymes in a recombinant form and study those protein kinases. Our results demonstrate that Serine/Threonine kinases CK1, CK2 and AUK ortholo-gues were able to phosphorylate Tcpolβ and those events have a positive influence on the enzyme activity, since phosphorylated Tcpolβ is greatly stimulated for DNA synthesis.

## Results

### *T. cruzi* DNA polymerase β contains multiple phosphorylation sites

Sequence analysis of Tcpolβ polypeptide indicated that contains different phosphorylation sites. Some of these sites have been previously described [17]. Interestingly, along the protein sequence multiple phosphorylation sites, which are the targets for Casein kinase 1 (CK1), Casein kinase 2 (CK2) and AUK, were identified. Fig 1 shows that using the online tools Net-Phos 3.1 and KinasePhos 2.0, set at high stringency, two CK1 target sites, two AUK target sites and six CK2 target sites were identified. We suggest that those potential phosphorylation tar-gets sites are important to regulate Tcpolβ activity and according to this end we have expressed

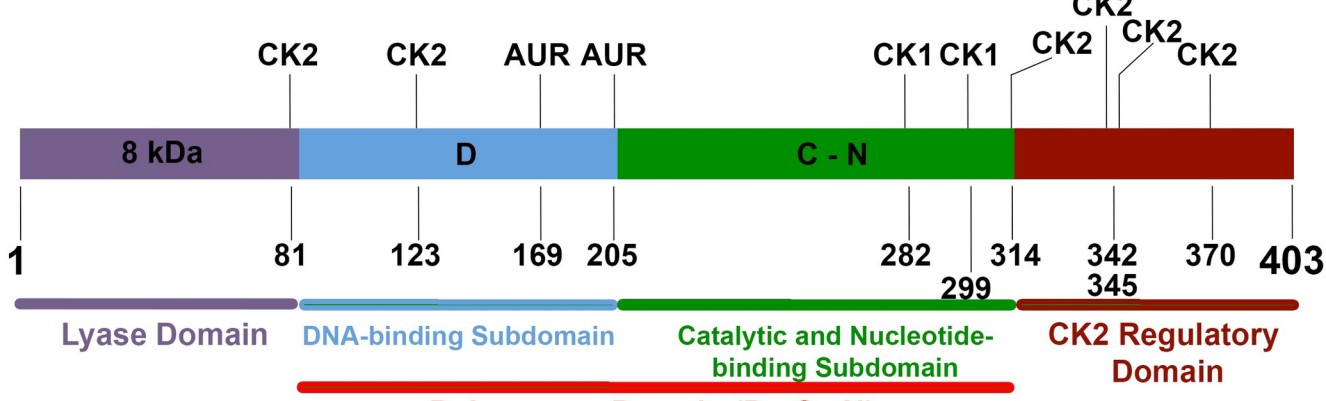

**Fig 1. Schematic representation of Tcpolβ.** A) The enzyme displays a three-domain structure with an 8 kDa N-terminal lyase domain (purple), a middle DNA-binding subdomain (D, sky blue), a catalytic and nucleotide-binding subdomain (C-N, green) and a putative C-terminal domain, which we named CK2 regulatory domain, since possesses four CK2 phosphorylation sites (brown). This extra C-terminal domain is absent in human DNA pol β. Also, the positions of the protein kinase phosphorylation sites are indicated. B) Phosphorylation sites by the three indicated protein kinases (sequence and position) are indicated. Note that those sites do not overlap.

recombinant CK1, CK2 and AUK protein kinases to evaluate Tcpolβ as substrate of those selected kinases. We have selected for those studies a functional orthologue of Aurore B kinase, which is named TcAUK1, however, we cannot rule out the possibility that TcAUK2 and TcAUK3 might also phosphorylate Tcpolβ, since genes encoding TcAUK2 and TcAUK3 exists in the *T.cruzi* genome (see discussion).

### Identification of CK1, CK2 and TcAUK1 encoding genes in the *T. cruzi* genome

To identify, clone, express and confirm that those kinases can phosphorylate recombinant Tcpolβ, we searched extensively the *T. cruzi* protein database using the gapped BLASTP program and as queries the kinase counterpart from *S. pombe* and *S. cerevisiae* and then compared

with the mammalian counterparts (S1–S4 Figs). In mammalians, the family of CK1 is composed by six members encoded by distinct genes (α, δ, ε, γ1, γ2 and γ3) [20] and they display the highest homology in their kinase domain (50–90% identical) with a lower degree of homology in the rest of the polypeptide. CK1 functions as a monomer and all the family members have similar substrate specificity. CK2 is a tetrameric enzyme which is composed most often by two catalytic subunits (α and α') encoded by separate genes and two regulatory subunits (β), encoded by a single gene. In mammals, AUK is a kinase family with three different members, AUKA, AUKB and AUKC encoded by distinct genes. We chose the AUKB-like protein for our further studies, since in higher eukaryotes this orthologue has several functions in mitotic spindle assembly and chromosome segregation [21], (see discussion).

## Cloning, expression and functional characterization of *T. cruzi* CK1

In mammals CK1α isoform is one of CK1 kinases that phosphorylates most of the consensus sites of the mammalian p53 RNA polymerase II transcriptional activator, which controls cell proliferation and the absence of this pathway is a common feature in all cancers where cell cycle is deregulated [22–24]. The relationship between CK1α kinase and p53 functions is the reason to select a *T. cruzi* CK1α-like orthologue to further studies. As described in S1 Fig, the alignment of CK1α amino acid sequences from mammals, fission yeast and *Leishmania* with the *T. cruzi* CK1α-like, displayed a high percentage of identity in the protein domains where the active site is located (S1 Fig). Also, the database search shows that there are different genes in the *T. cruzi* genome that encode several CK1 kinases (S5 Fig, see Discussion). The cDNA encoding this protein was synthetized at Genscript (NJ, USA) and recoded according to the codon usage of *E. coli* to obtain high levels of protein expression. This nucleotide sequence was inserted in a pET system, expressed in bacteria and the recombinant protein was purified and renatured for further functional analysis (see Methods). As displayed in Fig 2A, the purified renatured recombinant CK1α obtained was essentially pure as judged by SDS-PAGE followed by CBR-250 staining. The next step was the evaluation of the enzymatic activity of the purified protein kinase. To this end, a phosphorylation assay was performed using casein, which is the typical standard substrate for casein kinases. Fig 2B shows the successful phosphorylation of casein substrate using the recombinant *T. cruzi* CK1α. In addition, it can be observed the successful auto-phosphorylation of CK1α (Fig 2B), which is indicative of the functionality of the recombinant enzyme. Addition of the unspecific CK1 inhibitor heparin in the assays decreases the levels of phosphorylated casein in a dose-dependent manner (Fig 2B and 2C). Heparin inhibits casein phosphorylation by TcCK1 at concentration of 5 and 20 μg/ml however, it has no effect on the auto-phosphorylation activity of TcCK1. Afterwards, instead of casein, purified recombinant Tcpolβ was used as substrate. It can be observed that recombinant CK1α was able to phosphorylate Tcpolβ (Fig 2D and 2E), as it was predicted by previous bioinformatic analysis (see Fig 1 for consensus sites). The Tcpolβ substrate phosphorylation follows a dose-dependent manner (Fig 2D and 2E). Both Tcpolβ and CK1α have the same apparent molecular weight and cannot get separated by SDS-PAGE, however, the signal increasing observed in Fig 2D is due to the augment of Tcpolβ substrate, since the polymerase has no kinase activity and an unfolded inactive Tcpolβ is not a substrate for CK1α (S6 Fig). Therefore, we concluded that *T. cruzi* CK1α is capable to *in vitro* phosphorylate TcPolβ.

## Cloning, expression and functional characterization of *T.cruzi* CK2

Genes encoding CK2α and CK2β orthologues are present in the *T. cruzi* genome and the encoded polypeptides display high identity with the corresponding orthologues from fission yeast and human, mainly in the catalytic domain (active site and activation loop domains of

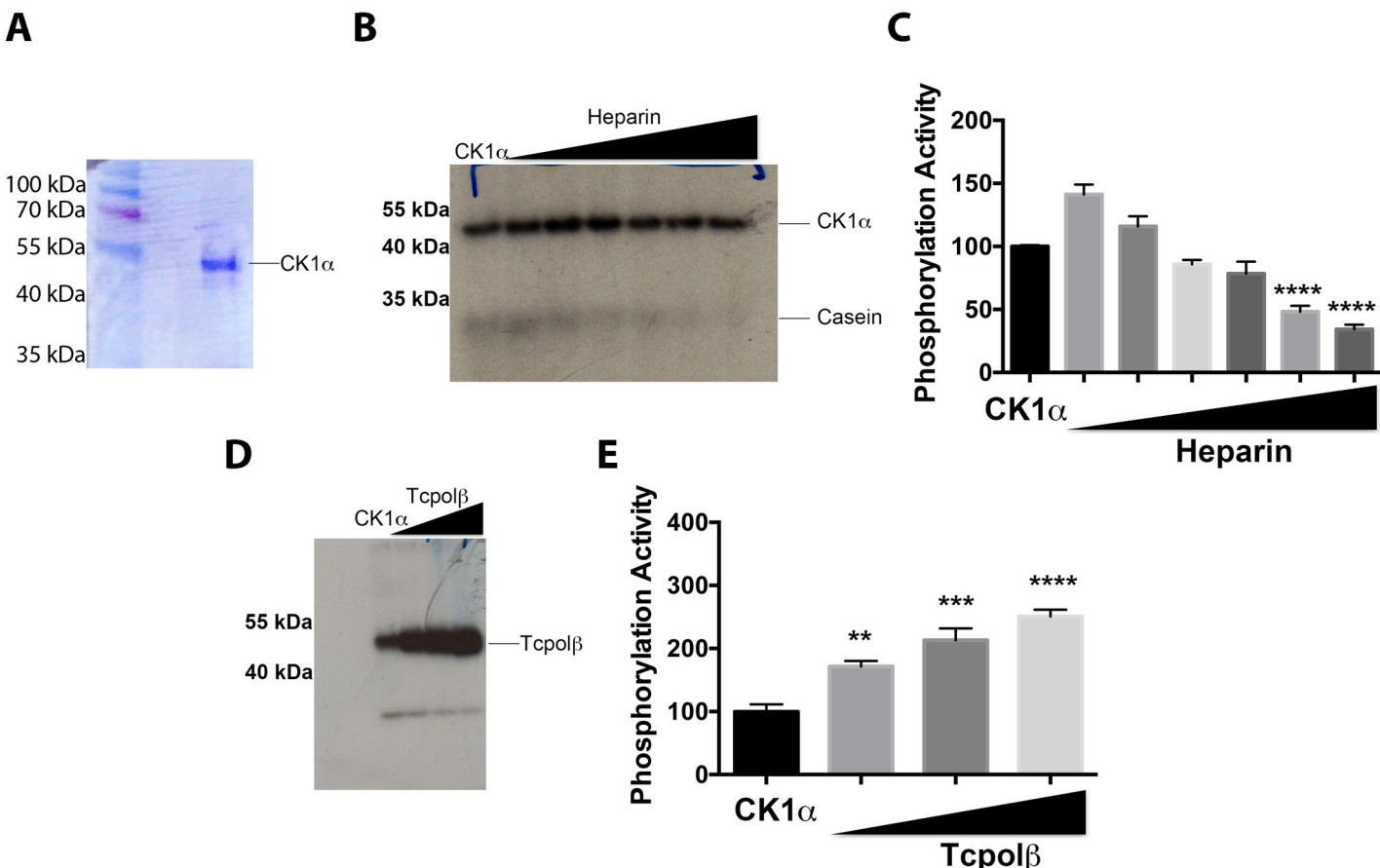

**Fig 2. A protein orthologous of human CK1α from *T. cruzi* has kinase activity.** A) SDS-PAGE analysis of the purified CK1α from *T. cruzi*. B) Casein phosphorylation by CK1α. The casein substrate (500 ng) was added in all experiments and from the second line heparin was added at concentrations of 0, 1, 2, 4, 10 and 20 μg/ml. It is observed that casein phosphorylation is strongly inhibited by heparin, however, the auto-phosphorylation of CK1α remains unaltered. C) Plot displaying the quantification of the inhibition of casein phosphorylation by heparin. D) Phosphorylation of Tcpolβ by CK1α. The first lane contains the phosphorylation mix and Tcpolβ. The following lanes contain 20 pmol of CK1α and where is indicated 50, 100 and 200 ng of Tcpolβ substrate. It can be observed that when increasing the substrate, the phosphorylation is higher, indicating that Tcpolβ gets phosphorylated. The faster migrating labeled polypeptide (NS) is a minor non-specific signal, which is present in the CK1α protein preparation. E) A plot that shows the quantification of the Tcpolβ phosphorylation by CK1α. Results are expressed in arbitrary units and represent the mean +/- SD of three independent experiments. Significance is defined as $p < 0.05$ when the mean of the experiments is compared with CK1α control. ** indicates $p < 0.01$; *** indicates $p < 0.001$ and **** indicates $p < 0.0001$.

CK2α, S2 and S3 Figs for CK2β). The cDNAs encoding both subunits were separately inserted in the pET system and expressed in *E. coli*. The CK2 holoenzyme was purified as described in Materials and Methods. Fractions from the column were analyzed by SDS-PAGE, followed by CBR-250 staining (Fig 3A, fractions 4 to 16). The TcCK2α subunit has an apparent MW of 40 kDa, whereas TcCK2β has an apparent MW of 50 kDa which is larger than predicted from its amino acid sequence (32 kDa). The reconstituted TcCK2 holoenzyme (Fig 3A, fraction 7) was used to test for kinase activity by using the classic standard casein substrate, however, we were unable to detect phosphorylation of this substrate. Next, we tested whether TcCK2α can phosphorylate Tcpolβ and whether TcCK2β has stimulatory effect on the phosphorylation activity of the TcCK2α catalytic subunit. As it is shown in Fig 3B and 3C, Tcpolβ is phosphorylated by TcCK2α at low levels, but those levels increased when increasing amounts of TcCK2β were added to the reaction mix. This result indicates that TcCK2β has a regulatory stimulatory activity on TcCK2α. A hybrid CK2 holoenzyme, in which TcCK2β was replaced by fission yeast CK2β (pCK2β; MW of 27 kDa), is also active and has higher activity than the TcCK2α/β

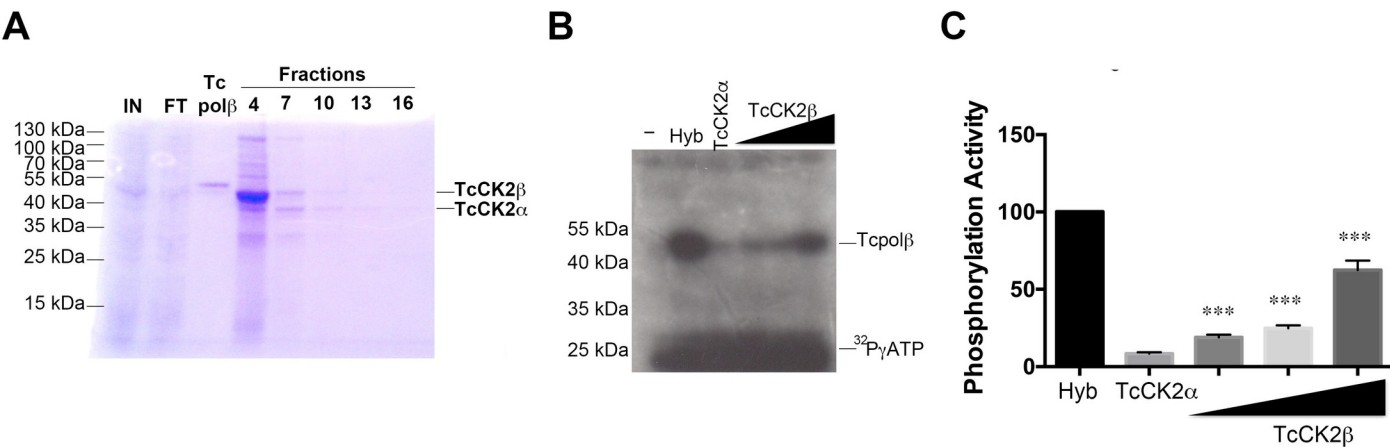

**Fig 3. TcCK2 is able to phosphorylate Tcpolβ.** A) Purification of TcCK2 holoenzyme and analysis by SDS-PAGE. Crude fractions of TcCK2α and TcCK2β were mixed and purified through NTA-Ni-Agarose as described in Materials and Methods. The gel shows the Tcpolβ substrate and the fractions eluted from the column. Fraction number 7 was used in the initial attempts to phosphorylate casein, however, those experiments failed. IN: input and FT: material that did not bind to the resin. B) TcCK2β can stimulate the activity of TcCK2α. In this experiment it can be observed that TcCK2α does not get auto-phosphorylated (-). Hybrid CK2 (Hyb) is able to phosphorylate Tcpolβ substrate and TcCK2α by itself can phosphorylate Tcpolβ at low levels, however, it is stimulated by TcCK2β, although at lower levels than pCK2β. In this experiment, 20 pmol of TcCK2α was used in each experiment with increasing amounts (8, 16 and 32 pmol) of TcCK2β as indicated in the figure. The heavy band on the bottom of film and labeled 32PγATP is free nucleotide that did not run out of the gel and covers phosphorylation of pCK2β. (-) indicates Tcpolβ substrate alone. C) Plot that shows the stimulation levels of TcCK2β on TcCK2α to phosphorylate Tcpolβ. Results are expressed in arbitrary units and represent the mean +/- SD of three independent experiments. Significance is defined as $p < 0.05$ when the mean of the experiments is compared with hybrid enzyme. *** indicates $p < 0.001$.

(Fig 3B and 3C; lane labelled Hyb). This is perhaps due to that pCK2β gets phosphorylated and it has a better stimulatory activity on TcCK2α (see below).

## Cloning, expression and functional characterization of TcAUK1

In order to analyze whether or no AUK could phosphorylate Tcpolβ, we decided to search for a *T. cruzi* AUKB orthologue, since this protein plays a role in mitosis in higher eukaryotes [21]. An AUKB-like protein was searched in the database by querying with the AUKB orthologue of fission yeast and human. The corresponding protein identified as TcAUK1 was compared with orthologues from fission yeast and human. In S4 Fig can be observed that TcAUK1 displays high identity with the corresponding orthologues, specifically in the segments where the active site and activation loop domain are located. The cDNA encoding TcAUK1 was inserted in a pET system and expressed in bacteria. The recombinant protein was renatured and purified (see Methods). The purity of the recombinant protein was analyzed by SDS-PAGE and it can be observed in Fig 4A. To evaluate the function of the recombinant enzyme, we checked the reported auto-phosphorylation ability of AUK from human. As reported in Fig 4B and 4C, the phosphorylated protein observed was the recombinant TcAUK1, indicating that the kinase is functional and meets the criteria of AUKB. However, when Tcpolβ was added as a substrate to the assay, a strong phosphorylation signal was obtained and the phosphorylation level was dependent on the amount of Tcpolβ substrate added to the assay (Fig 4B and 4C). Due to the fact that TcAUK1 and TcPolβ migrates close together in a SDS-PAGE, we performed a phosphorylation assay with unfolded inactive Tcpolβ, which is not a substrate for CK1α, and we observed that the addition of the inactive polymerase does not augment the phosphorylation signal (S7 Fig), indicating that the signal increment in Fig 4B is not due to stimulation of the TcAUK1 autophosphorylation activity, and also suggests that only properly folded and active Tcpolβ is a substrate for protein kinases. Thus, we concluded from those analysis that Tcpolβ is a substrate for TcAUK1.

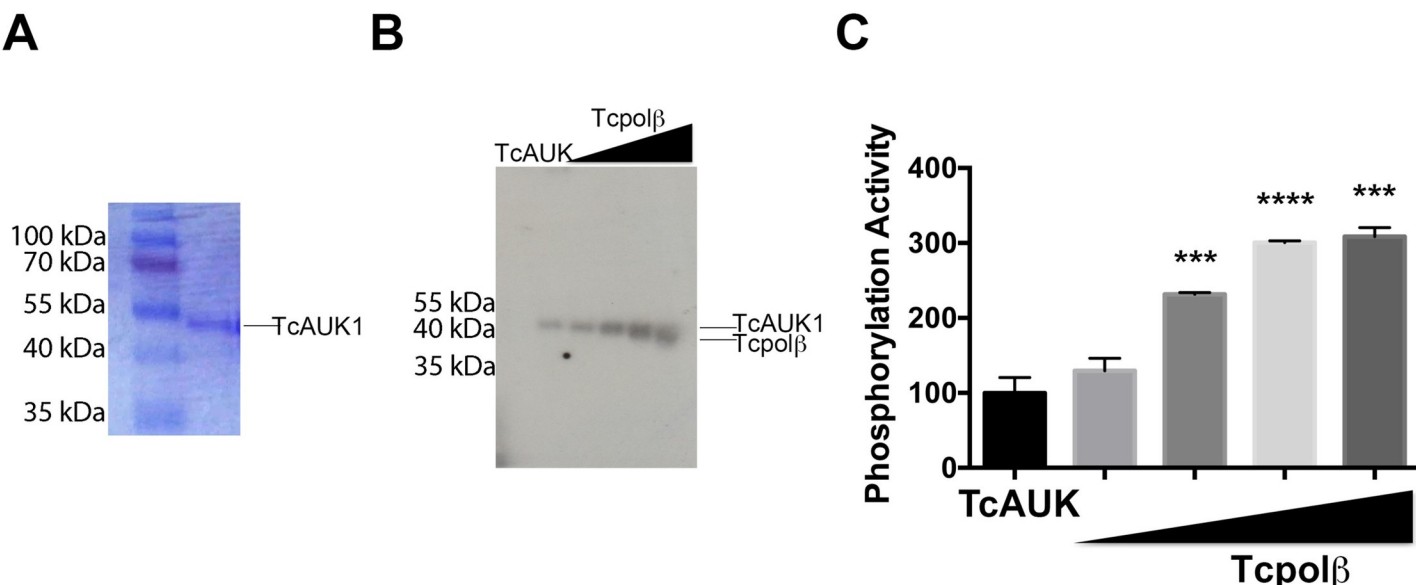

**Fig 4. Tcpolβ is a phosphorylation substrate for TcAUK1.** A) SDS-PAGE analysis of purified TcAUK1. B) Tcpolβ is phosphorylated by TcAUK1. It is observed that TcAUK1 (20 pmol) is auto-phosphorylated and it can also phosphorylate Tcpolβ, since increasing amounts (0, 25, 50 and 100 ng) of Tcpolβ is added a higher phosphorylation signal is obtained and also in the last lane a new phosphorylated protein band appeared, which is indicated in the figure as Tcpolβ. However, it might be also possible that the polypeptide doublet corresponds to two species of phosphorylated Tcpolβ, as this DNA polymerase contains two AUK phosphorylation consensus sites. C) Quantification plot of Tcpolβ phosphorylation by TcAUK1. Results are expressed in arbitrary units and represent the mean +/- SD of three independent experiments. Significance is defined as $p < 0.05$ when the mean of the experiments is compared with TcAUK1 assay control. *** indicates $p < 0.001$ and **** indicates $p < 0.0001$.

## Effect of pharmacological inhibitors on the activity of TcCK2, TcCK1 and TcAUK1

In order to complement the characterization of each recombinant protein kinase, we used specific inhibitors for each of those protein kinases, previously described to inhibit mammalian enzymes. We performed phosphorylation assays in the presence of different inhibitors specific to each protein kinase. Heparin, a well-characterized mammalian CK1/CK2 inhibitor was used to inhibit the casein phosphorylation by CK1α. Those results were described in Fig 2A and 2B in a previous section. To inhibit Tcpolβ phosphorylation by CK2, we used TBB (4,5,6,7-tetrabromobenzotriazole) a specific ATP/GTP competitive mammalian CK2 inhibitor. Also, D4476 (CK1 inhibitor) and SNS314 (AUK inhibitor) were used to characterize those protein kinases further.

In order to test the specific mammalian CK2 inhibitor, namely TBB, we used the hybrid CK2 holoenzyme in which TcCK2β was replaced by pCK2β, which possesses stimulatory activity and several CK2 phosphorylation sites in the first fourteen amino acid residues of the N-terminus of the polypeptide (S3 Fig). As it can be seen in Fig 5A, TcCK2α is capable to phosphorylate pCK2β and this phosphorylation is inhibited by TBB (lower band). Quantification of the inhibition (Fig 5B) indicates that pCK2β phosphorylation by TcCK2α is inhibited by TBB with an $IC_{50}$ of 0.5–1.0 μM in agreement with the $IC_{50}$ reported for rat and human recombinant CK2 (0.9 μM and 1.6 μM respectively). These results indicate that TcCK2 might have a slightly different preference of substrate that human recombinant CK2. The observation that pCK2β can functionally replace TcCK2β underscores the functional and structural conservation of the CK2β subunits through the evolution. The protein kinase activity is specific of the TcCK2α subunit, since pCK2β has no protein kinase activity by itself and exclusively serves as stimulatory subunit (S8 Fig). On the other hand, TBB does not inhibit Tcpolβ

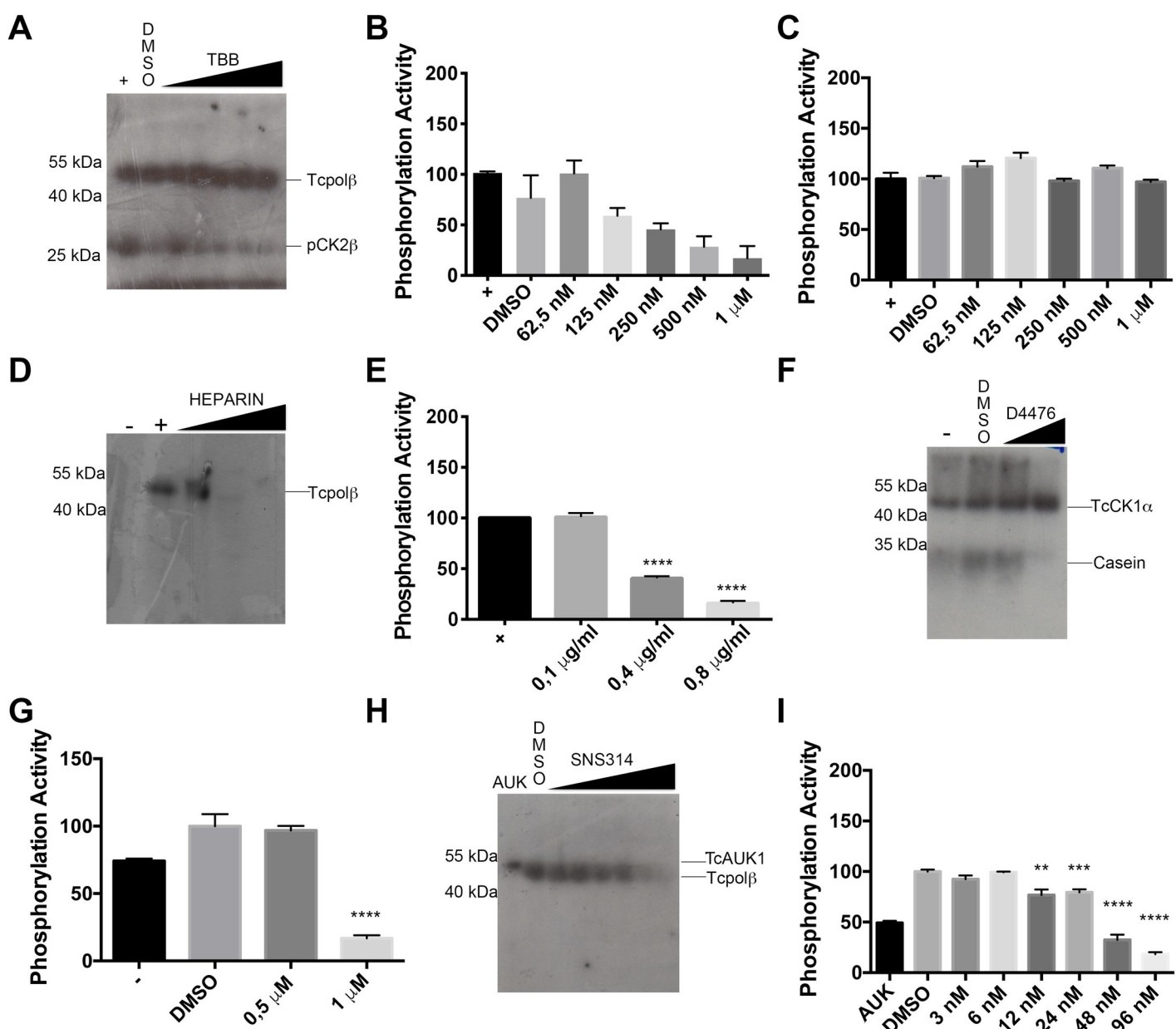

**Fig 5. The effects of different specific inhibitors on the kinase activity of the studied *T. cruzi* protein kinases.** A) A hybrid CK2 holoenzyme can phosphorylate Tcpolβ. In this experiment, 20 pmol of TcCK2α was mixed with 15 pmol of *S. pombe* CK2β (pCK2β) and used to phosphorylate 200 ng of Tcpolβ substrate. It can be observed a strong phosphorylation of Tcpolβ (upper band) as well as of pCK2β (lower band). Also, TBB was added to the assays at concentrations of 62.5 nM, 125 nM, 250 nM, 500 nM and 1 μM. TBB concentrations of 500 nM and 1 μM can inhibit the phosphorylation of pCK2β, however, is unable to inhibit the phosphorylation of Tcpolβ substrate. (+) indicates complete reaction with hybrid holoenzyme and Tcpolβ substrate. B) Plot that show the effect of TBB on pCK2β phosphorylation. Results are expressed in arbitrary units and represent the mean +/- SD of three independent experiments. (+) indicates complete reaction with hybrid holoenzyme and Tcpolβ substrate. C) Quantification of the inhibition activity of TBB on the phosphorylation of Tcpolβ. D) Heparin, which is a known inhibitor of mammalian CK2, can inhibit the kinase activity of TcCK2 holoenzyme. We used concentrations 0.1 μg/ml, 0.4 μg/ml and 0.8 μg/ml of heparin. In this experiment we used the holoenzyme from fraction 7 in Fig 3A. CK2 holoenzyme does not get auto-phosphorylated (-), however, is able to phosphorylate Tcpolβ (+) and the phosphorylation is completely inhibited by heparin at 0.8 μg/ml. E) Plot that shows the quantification of (D). F) The CK1 kinase inhibitor D4476 was used to inhibit the kinase activity at 0.5 and 1 μM. This inhibitor at the concentration of 1 μM was able to inhibit CK1α kinase activity towards casein, however, it was unable to inhibit the auto-phosphorylation of CK1α. G) Plot that shows the quantification of (E). H) The AUK inhibitor SNS314 was able to inhibit the activity of TcAUK1. I) It can be observed that this inhibitor can inhibit the activity of this kinase at 48 nM as indicated in the plot. Results are expressed in arbitrary units and represent the mean +/- SD of three independent experiments. Significance is defined as $p < 0.05$ when the mean of the experiments is compared with their respective control assay. ** indicates $p < 0.01$, *** indicates $p < 0.001$ and **** indicates $p < 0.0001$.

phosphorylation by TcCK2α (Fig 5A and 5C). Heparin, a well characterized CK2 inhibitor, was able to inhibit the phosphorylation of Tcpolβ by TcCK2 at 0.8 μg/ml, which is at least 6-fold higher than the concentration that inhibits fission yeast and mammalian CK2 (Fig 5D and 5E). The presented results indicate that TcCK2 is a bona fide CK2 kinase.

The specific CK1 inhibitor D4476 decrease the levels of phosphorylated casein but not the levels of auto phosphorylated *T. cruzi* CK1α (Fig 5F and 5G), as it was observed earlier with heparin. D4476 inhibits mammalian CK1δ with an $IC_{50}$ of 0.3 μM, which is lower compared to TcCK1 described in this work ($IC_{50}$ close to 0.8 μM). Evaluations using Tcpolβ were not done, because both proteins TcCK1α and Tcpolβ migrate close together in SDS-PAGE. To inhibit the activity of TcAUK1, we used SNS314 and Tcpolβ as substrate. A decrease of over 20% was observed with concentrations of inhibitor of 12 and 24 nM (Fig 5H and 5I), whereas higher concentrations (48 and 96 nM) decreased over 60% the phosphorylation activity of TcAUK1. Those inhibitor levels are in agreement with those reported for mammalian AUKB ($IC_{50}$ = 31 nM).

### The activity of Tcpolβ increases by phosphorylation by TcCK1, TcCK2 and TcAUK1

To evaluate the effect of phosphorylation on the activity of Tcpolβ, we used qualitative assays and a protocol in which suboptimal concentrations of Tcpolβ (which do not produce detectable synthesis) were incubated with increasing amounts of protein kinases and the polymerase activity was measured by two independent assays. First, a scintillation liquid assay, where we previously phosphorylated Tcpolβ by each kinase, followed by the assay on activated calf thymus DNA. Afterwards, the incorporation of the isotope on the newly synthetized DNA was determined on a scintillation counter. Second, we previously phosphorylated Tcpolβ by the kinases and then we assayed the activity on SDS-PAGE gels containing activated calf thymus DNA. Results are presented in Fig 6A (scintillation), 6B and 6C (activity SDS-PAGE). It can be observed that assays containing Tcpolβ plus protein kinase, in the absence of ATP, have no

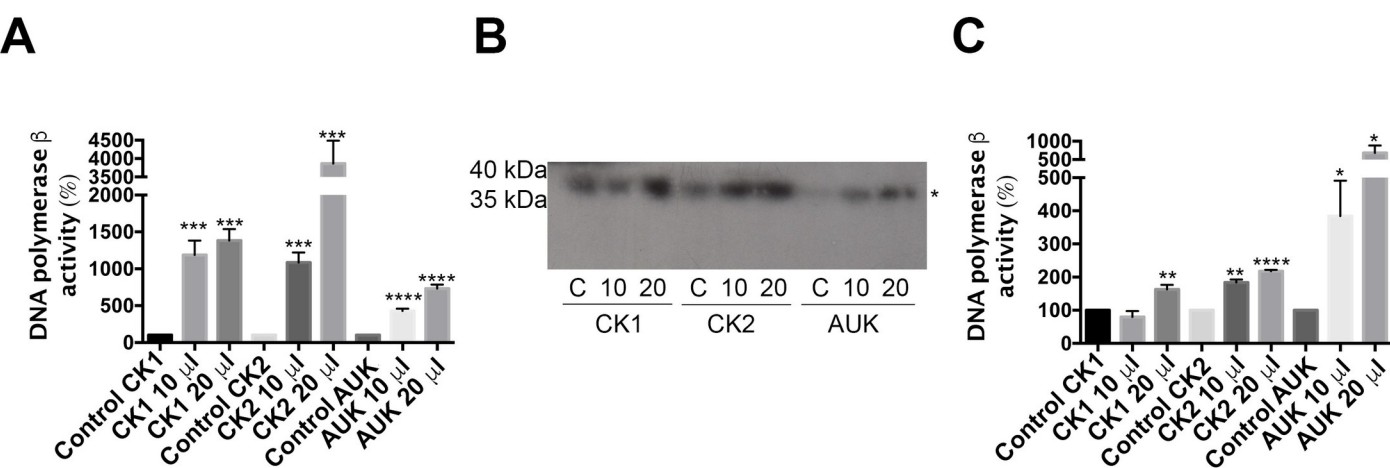

**Fig 6. The phosphorylation state of Tcpolβ by the studied protein kinases increases its polymerizing activity.** A) Liquid scintillation assay. In this experiment, 20 ng of Tcpolβ were incubated with the indicated kinases, without ATP for controls (with 20 pmol of each protein kinase), and with 10 and 20 pmol of each indicated kinase enzyme. After 30 minutes the reaction was spotted in a DE81 paper piece and processed as indicated in Materials and Methods. It can be observed a high stimulation of the polymerizing activity of the phosphorylated Tcpolβ. B) To further confirm this activity an identical experiment as in A was done, but the reaction was loaded in an activity SDS-PAGE gel. The results are showed and is observed that CK1, CK2 and TcAUK1 are capable to stimulate the polymerizing activity of Tcpolβ. C) Quantification plot of the experiment in B). Results are expressed in arbitrary units and represent the mean +/- SD of three independent experiments. Significance is defined as $p < 0.05$ when the mean of the experiments is compared with the respective control assay (CK1, CK2 and TcAUK1). * indicates $p < 0.05$, ** indicates $p < 0.01$, *** indicates $p < 0.001$ and **** indicates $p < 0.0001$.

detectable polymerase activity, however, after phosphorylation the enzyme had considerable DNA polymerase activity. The effect of CK1, on the DNA polymerase activity, was a 10–14-fold increment using the scintillation assay and 2-fold increment using the SDS-PAGE activity. As for the TcCK2 phosphorylation effect on Tcpolβ activity was 40-fold increment using the first approach and 2.5-fold using the SDS-PAGE activity. TcAUK1 phosphorylation effect on Tcpolβ was 5–8-fold increment using the scintillation assay and 8-fold increment using the second approach. By using both approaches, it was observed a positive effect of phosphorylation on Tcpolβ DNA polymerase activity, although, the extend of the effect is higher using the scintillation assay. This is due to the nature of each assay, since in the second assay the polymerase must be renatured on the gel. We can conclude that the phosphorylation of Tcpolβ by those three different kinases has a positive effect on the DNA polymerase activity and we suggest this modification might have a functional role *in vivo*.

## Discussion

The DNA repair systems are essential to maintain genome integrity, therefore can avoid mutations to ensure cell survival. DNA lesions frequently generated such as abasic sites, DNA strand breaks, oxidized, deaminated and alkylated are repaired mainly by the base excision repair system (BER). The BER starts with a glycosylase which removes the damaged base, producing an apurinic/apyrimidinic site (AP site) followed by an incision of the phosphodiester backbone 5′ to the abasic site by an AP endonuclease [25]. This step leaves a single-nucleotide gap with 3′-hydroxyl and 5′-deoxyribose phosphate at the gap margins. After, DNA polymerase β incorporates the missing nucleotide according to the template instructions and removes the 5'- deoxyribose-5- phosphate (intrinsic dRP lyase activity) and the chain is sealed by a ligase [25,26]. In mammals, DNA polymerase β is the main polymerase involved in the BER repair system and carries out the repair of 8-oxoG lesions and other oxidized pyrimidines in the nuclear genome. However, in *T. cruzi* this DNA polymerase (Tcpolβ) is located into the mitochondrion and plays a role in the base excision repair (BER) system [18,19]. Moreover, a second related DNA polymerase β is found into the mitochondrion and it is named Tcpolβ-PAK, which is competent for *in vitro* trans-lesion synthesis (TLS) through 8-oxoG lesions and possesses dRP lyase activity [18]. Tcpolβ has DNA polymerase functions and an intrinsic dRP lyase activity, but it is unable to repair *in vitro* across 8-oxoG lesions by TLS [14]. Available evidence indicates that Tcpolβ repairs oxidative damage through BER of kinetoplast DNA (kDNA) and is also involved in its replication [18,27]. The overexpression of Tcpolβ renders the cells more resistant to oxidative damage by ROS and repair 8-oxoG lesions with a high efficiency compared to control cells. Interestingly, Tcpolβ has been purified from cell extracts as a doublet and only the larger form contains DNA polymerase activity [28]. *In vivo* Tcpolβ exists in two forms, a heavy (H) and a light (L) one. The H form is highly phosphorylated and active in DNA synthesis, whereas the L form is almost inactive [19]. Moreover, cells under ROS treatment respond overexpressing Tcpolβ, specially the active H phosphorylated form [19]. However, the protein kinases that phosphorylate Tcpolβ are still unknown.

In attempts to investigate the putative protein kinases that phosphorylates Tcpolβ, we searched potential phosphorylation sites and we found that this enzyme has consensus sites for three kinases, namely CK1, CK2 and AUK. We searched for *T. cruzi* orthologues of those proteins at the protein database of the NCBI. We found several genes encoding CK1 (S1 Fig), an orthologue of CK2α, CK2β and three orthologues of AUK (TcAUK1-3). We selected the orthologues based mainly in the criteria of identity to orthologues from fission yeast and mammals. All three recombinant kinases were able to *in vitro* phosphorylate Tcpolβ. Moreover, the phosphorylation event stimulates the DNA polymerase activity and DNA synthesis ability of

Tcpolβ. Whether or not, the phosphorylation of Tcpolβ stimulates the intrinsic dRP lyase activity is still unknown. However, we believe that dRP lyase activity might be also stimulated by phosphorylation as well as its processivity and processive search of the DNA. Also, the phosphorylation might cause changes in the subcellular location of Tcpolβ, since it has been reported that in hydrogen peroxide treated epimastigote cells, Tcpolβ appears in new additional focus outside of the kDNA as compared to untreated cells [18]. It has been recently shown that mammalian DNA polymerase β can perform a processive search (DNA scanning) looking for DNA damage using its lyase domain [29,30]. This indicates that the dRP lyase domain of DNA polymerase β can non-specifically interact with DNA during the scanning process. Tcpolβ might perform a similar function, since it has an active dRP lyase domain and can be crosslinked to kDNA in the absence of DNA damage [19].

CK1 is a highly conserved protein kinase family from protozoa to mammals. Its members are ubiquitously expressed and have pleiotropic effects in the cell. The CK1 family play key roles in many cellular processes including cell proliferation and differentiation, apoptosis, vesicular trafficking, DNA repair, mRNA processing, cytoskeleton dynamics, and circadian rhythms [31–33] On the other hand, CK2 is also a ubiquitously expressed pleiotropic protein kinase and play many major regulatory roles in several cellular processes including cell cycle, transcription, protein stability and degradation, translation, circadian rhythms, tumor progression in mammals and cell survival. CK2. It is ubiquitously expressed and has a great number of substrates. CK2 is highly conserved and broadly distributed across the eukaryotic kingdom, since can be found from protozoa to mammals [34–36] AUK is a small highly conserved family of protein kinases, which has three members (A, B and C) and are present from protists to mammals. AUK kinases play a major role in cell division via regulating mitosis especially the process of chromosomal segregation, and also have been implicated in regulating meiosis [21,37,38].

Mammalian DNA polymerase β has several consensus phosphorylation sites for several protein kinases including PKA, PKC, CK1, CK2, DNAPK, cdc2 and several others, but it has no consensus phosphorylation sites for AUK (NetPhos Server) [13], however, those consensus sites are not located in similar positions as those in Tcpolβ. CK1 and AUK phosphorylation sites are located at the polymerase domain of Tcpolβ, whereas one CK2 site is located at the limit in between the lyase and the polymerase domain and another one is present in the polymerase domain, however, four of those sites locate at the C-terminal extra domain, that we named CK2 regulatory domain, since it could be involved in regulating the subcellular location of the enzyme. The CK2 phosphorylation site located at the lyase domain might regulates the activity of this domain, since it is close to the lyase catalytic center. The AUK phosphorylation sites are located at the polymerase domain, specifically at the DNA-binding subdomain. We propose that CK1, CK2 and TcAUK1 could control the cell cycle, differentiation and DNA repair in *T. cruzi* through the regulation of key proteins involved in those processes. One of those proteins might be Tcpolβ, as this DNA polymerase is involved in BER repair system and kDNA replication.

With the advent of powerful NGS techniques, it has been possible to sequence and assemble the genome of several *T. cruzi* strains and reveals a greater complexity than it was believed earlier. Several multicopy gene families have been expanded in different strains [39,40]. One of those is the CK1 kinase gene family, which has been found in tandem repeats inside the longest tandem chromosomic repeat element [39]. S5 Fig shows a schematic alignment of the polypeptides encoded by those genes. Several of them show similar domains, other have extra domains and many of them lack one or more domains. It is unknown whether are all expressed, and many of those smaller ones cannot be functionals at all. Since CK1 is an important protein

kinase for cell function its expansion could be related with the survival and virulence of each *T. cruzi* strain. Studies on this issue deserves further efforts.

An important question is about the cellular signaling pathways which regulate trypanosomatid biological processes, such replication, transcription, cell division, differentiation and DNA repair. Furthermore, trypanosomatids should respond and adapt quickly to a series of extracellular and intracellular signals. In metazoan and fungi, the final targets of many protein kinases signaling cascades are transcription factors which are able to trigger the expression of a set of genes involved in that specific pathway. However, trypanosomatids are unable to regulate gene expression at the transcriptional level, since genes are indiscriminately transcribed in large polycistronic units. Therefore, signaling cascades in trypanosomatids must function at the post-transcriptional regulatory level and protein kinases should play a fundamental role in the regulation of trypanosomatid biological processes. Phosphorylation and dephosphorylation of downstream key molecules, which can regulate cell-specific and cell developmental processes have to be essential for trypanosomatids. Protein kinases could be key regulators of proteins involved in DNA replication, mRNA splicing, specific mRNA turnover, mRNA translation, DNA repair, cell cycle and many other trypanosomatid biological processes. Bioinformatic searches [41] for protein kinases (kinome) in the genome of three pathogenic trypanosomatids, namely *T. cruzi*, *Leishmania major* and *Trypanosome brucei*, have revealed that those parasites contain several groups of protein kinases that can play key roles in trypanosomatid biological processes. As an example, oxidative stress by hydrogen peroxide induces Tcpolβ overexpression and phosphorylation in *T. cruzi* cells [19]. The overexpressed Tcpolβ H form is highly phosphorylated *in vivo* and it is more active in DNA synthesis than the L form, which is dephosphorylated [19]. We speculate that ROS triggers a signal transduction pathway to signal protein kinases, which in turn can augment the translation and phosphorylation of Tcpolβ to potentiate its polymerase activity and in such a way can perform DNA repair. Those protein kinases could be CK1, CK2, AUK and perhaps MAPKs as well, since Tcpolβ is an *in vitro* substrate for CK1, CK2 and TcAUK1 and its phosphorylation by those protein kinases has a positive effect on DNA synthesis. Indeed, in other trypanosomatid, namely *Leishmania braziliensis*, has ectopic CK2 and secretes this protein kinase to the media, where mediates the association of macrophages and parasites, enhancing the virulence of CK2-secreting strains [42]. Also, *Leishmania tropica* can secrete constitutively CK1 and CK2 to the culture media and those protein kinases appear to be involved in promastigote morphology, cell growth and infectivity and the presence of those protein kinases might be fundamental for a successful parasite infection [43]. This initial hypothesis is outlined in Fig 7 and it is currently being tested in our laboratory.

In a very recent study, a TcAUK1 gene has been identified and characterized. This is an identical gene that encodes the protein studied in the present work [44]. TcAUK1protein has a similar location as mammalian AUKB, since it locates inside the nucleus and it is associated with the mitotic spindle during mitosis [44]. TcAUK1 is closely related to *Trypanosoma brucei* Aurora-B kinase (TbAUK1), which has a role in spindle formation, mitosis, cytokinesis, and organelle replication [45]. TcAUK1 protein has a changing location during the cell cycle of the parasite. At the interphase is into the kinetoplast and in mitosis is inside the cell nucleus. Its overexpression delays the G2/M inter-phase and this is due to a retarded beginning of kinetoplast duplication [44]. During interphase, TcAUK1 locates at the extremes of the kinetoplast, however, when it is overexpressed locates inside the nucleus throughout the entire cell cycle [44]. TcAUK1 seems to play a role during the initiation of kinetoplast duplication and perhaps carries out this role by direct phosphorylation of Tcpolβ, which is involved in kDNA repair and replication. Ours results support this notion, since TcAUK1 phosphorylates Tcpolβ *in vitro* and this polymerase is kinetoplast-located and participates in kDNA repair and

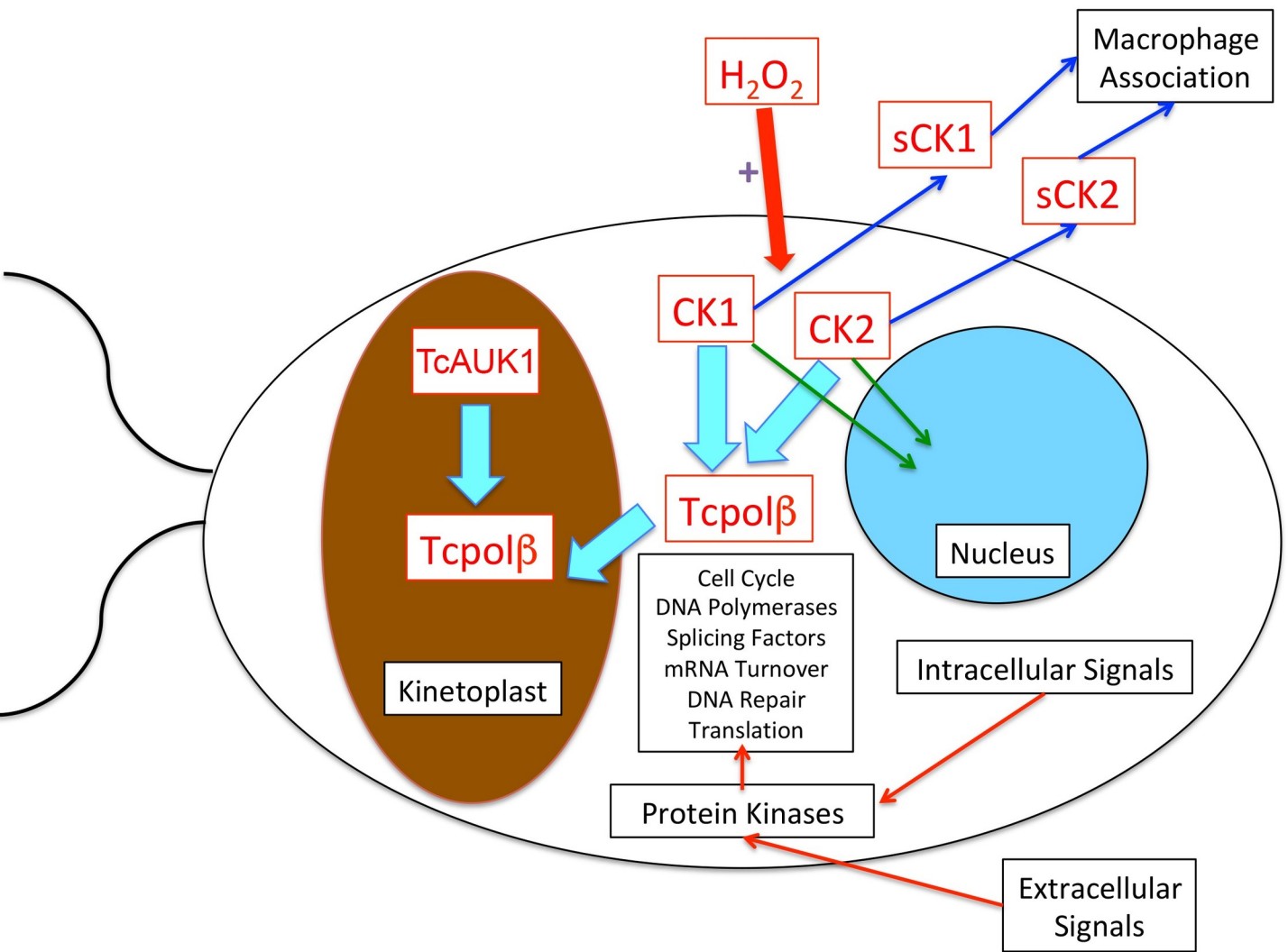

**Fig 7. Hypothesis on signal transduction in trypanosomatids.** Extracellular and/or intracellular signals can trigger protein kinase cascades, which can modify key downstream regulatory or effector molecules involved in DNA replication, mRNA splicing, mRNA turnover, mRNA translation, DNA repair, cell cycle and other fundamental biological processes. Trypanosomatids must heavily rely on phosphorylation of those key downstream effector molecules to regulate their biological processes. As an example, in *T. cruzi*, the ROS signal could be sensed on the membrane or travels the cytoplasmic membrane and then start a signal transduction pathway, involving CK1 and/or CK2, which can phosphorylate Tcpolβ modifying its DNA synthesis activity. Parasite secreted CK1 and CK2 (sCK1; sCK2) can enhance the virulence of some trypanosomatid strains, as they stimulate the association of macrophage and parasite cells. This hypothesis is easily testable by the use of well characterized pharmacological inhibitors to test their effects on the protein kinases and then on trypanosomatid biological processes. Alternatively, genetic approaches can be used to assay gene function, such as knockdown by RNA interference or gene knockout by CRISP/Cas-9 approaches. TcAUK1 has been located inside the mitochondrion together with Tcpolβ and it could directly modify its polymerase activity.

replication [15]. Interestingly, the *T. cruzi* genome encodes at least another two different AUKs, namely TcAUK2 and TcAUK3, which could play key roles in cell division [44].

The results presented here provide a framework to use known protein kinase inhibitors on *T. cruzi* cells and test any effects on Tcpolβ phosphorylation and cell proliferation. Also, the use of TcAUK1 inhibitors could affect Tcpolβ function and kDNA repair and replication. Since protein kinases are significantly different in this parasite compared to protein kinases from mammals, therefore, it is possible to search or design protein kinase inhibitors able to target the parasite protein kinase without harming the host. Some of those experiments are currently being carrying out in our laboratory.

## Methods

### Protein expression and purification

The ORFs of *T. cruzi* CK1 (RNF06843.1), *T. cruzi* CK2 (CK2α: XP_821063.1; CK2β: PWV06843.1) and TcAUK1 (ACA50091.1) were recoded according to the codon usage of *E. coli* and synthetized at Genscript (NJ, USA). After synthesis the cDNAs were inserted in frame in pET15b and BL21(DE3) cells were transformed. Cells were plated in Luria-Bertoni broth plates and a single colony was grown at 37˚C in Terrific Broth medium supplemented with ampicillin (0.1 mg/ml). Cultures were grown to O.D.$_{600nm}$ = 0.8–1.0. Then, protein expression was induced by the addition of 0.5 mM IPTG to each culture. After 3 hours, cells were pelleted and collected material was processed and the recombinant protein was purified according to Maldonado *et al* 2015 [17] and tested by enzymatic activity. To reconstitute the *T. cruzi* CK2 holoenzyme both subunits were mixed at 1:1 molar ratio, dialyzed against folding buffer to eliminate the guanidine hydrochloride that was used to solubilize the proteins from the bacterial inclusion bodies. After dialysis, the proteins were allowed to get the proper folding, loaded on a NTA-Ni-Agarose column and eluted with an Imidazole gradient from 5–250 mM. Protein purity was determined by 10% SDS-PAGE. Also, pure recombinant proteins were analyzed by MALDI-TOF-TOF to confirm its identity. The mass spectroscopy analyses were done at the Institute Pasteur (Montevideo, Uruguay).

### Phosphorylation assays

The assay for CK1 was done in buffer A (25 mM HEPES pH 7.8, 5%v/v glycerol, 10 mM KCl, 0.1 mM EDTA, 5 mM MgCl2, 5 mM DTT, O.01% v/v NP-40, 0.05 mM ATP and 4 μCi γ$^{32}$P ATP). For CK2 was done in buffer B (50 mM HEPES pH 7.8, 5% v/v glycerol, 5 mM KCl, 0.1 mM EDTA, 5 mM DTT, 0.01% v/v NP-40, 0.05 mM ATP and 4 μCi γ$^{32}$P ATP) and for TcAUK1 was performed in buffer C (50 mM HEPES pH 8.0, 5% glycerol, 10 mM KCl, 0.1 mM EDTA, 5 mM MgCl$_2$, 5 mM DTT, 0.1% NP-40, 0.05 mM ATP and 4 μCi γ$^{32}$P ATP). All reactions were done in 25 μl final volume. Amounts of each kinase and substrate were as indicated in each figure legend. For inhibitor assays, increasing amounts of heparin (0 to 20 μg/ml), D4476 (0.5 and 1 μM; TOCRIS, USA), SNS314 (3 nM to 96 nM; TOCRIS, USA) and TBB (62.5 μM to 1 mM; TOCRIS, USA) were used in the experiments. The reactions were performed at 24˚C for 30 minutes and terminated by addition of SDS-PAGE sample buffer. Samples were heated at 95˚C and loaded on a 10% SDS-PAGE gel. After the electrophoresis was completed, the gels were exposed overnight to X-ray films. The films were processed, scanned and signals quantified using Image J software (NCBI, USA). Briefly, pixels from every film band associated to phosphorylation were determined using the tool "gel analysis" from the software. Background was subtracted to eliminate the noise and over exposition. All experimental data were normalized with the corresponding control according to each experiment. After normalization, data were expressed as percentage of control (arbitrary units) and indicated in figures as "phosphorylation activity".

### Liquid scintillation DNA polymerase assay

Tcpolβ was phosphorylated by CK1, CK2 and TcAUK1 in the above buffers supplemented with 1 mM ATP. Considering that the amount of Tcpolβ to saturate the enzymatic activity in the assay was 40 pmol, we set up the reactions with 1/5 of the maximal level (8 pmol). In this way we were able to record any changes in Tcpolβ activity. Phosphorylation reactions were done for 30 minutes at 24˚C. Afterwards, Tcpolβ reaction buffer (40 mM Tris-HCl pH 8.0, 10 mM DTT, 1 mM MnCl2, 40 mM KCl, 1 mM EGTA, 0.2 mg/ml BSA, 40 mg/ml DNAseI-

activated calf thymus DNA, 500 μM dNTP mix (dCTP, dGTP and dTTP, and 5 μM dATP) was added together with 3.32 nM α$^{32}$P dATP. (3000 Ci/mmol, Perkin Elmer, USA) and incubated for additional 30 minutes. Reactions were terminated by spotting each reaction onto a DE81 paper piece (Whatman, USA). Paper pieces were washed three times with 0.3 M Na$_2$HPO$_4$ to eliminate non-incorporated radionucleotides. Then, paper pieces were dried and radionucleotide incorporation was measured in a scintillation counter. Collected data was plotted the Tcpolβ activity was expressed as percentage of activity where 100% is Tcpolβ activity without protein kinases. Data were normalized with the corresponding control (CK1, CK2 or TcAUK1).

### DNA polymerase activity gel

The activity of Tcpolβ was detected in a SDS-PAGE according to literature [19], with slight modifications. Briefly, Tcpolβ was phosphorylated with the different protein kinases using the above buffer supplemented with 0.4 mM ATP for 20 minutes at room temperature, then was mixed with calf serum (10% v/v final concentration) and Laemmli SDS-PAGE sample buffer containing 10 mM DTT. Then, samples were heated at 37˚C for 5 minutes and loaded onto 8.5% SDS-PAGE gel, which was supplemented with 100 μg/ml of activated calf thymus DNA. When run was complete, the gel was washed twice with 50 mM Tris-HCl pH 8.0. Then, the gel was incubated with folding buffer (50 mM Tris-HCl pH 8.0, 0.5 mM EDTA, 5 mM DTT, 0.5 mg/ml BSA, 15% v/v glycerol, 0.01% v/v NP-40 and 4 mM Mg-acetate) for 3 hours at 24˚C. The gel was incubated overnight at 4˚C with fresh folding buffer. After this step, the gel was incubated with fresh folding buffer supplemented with 250 mM KCl, 25 μM dTTP, dGTP, dCTP, 1 μM dATP, 1 μl/ml α$^{32}$P dATP (3000 μCi/mmol) and 3 mM MnCl$_2$. Incubation was performed with gently agitation overnight at room temperature. Then, the gel was washed five times with 5% v/v TCA and 2% w/v potassium pyrophosphate for 30 minutes (each wash) at room temperature. A last wash was performed at 50˚C for 45 minutes. After the last step, the gel was exposed to X-ray films. The films were processed, scanned and signals quantified using Image J software as previously described.

### Statistical analysis

Differences between means in all data presented in this work were analyzed for statistical significance using Student's t-tests in the Prism 6,0 software (GraphPad, USA). Significance was considered when $p < 0.05$. Each experiment was repeated at least three times.

### Supporting information

**S1 Fig. Multiple alignment sequence of CK1 orthologous from different species.** Sequences were aligned using the ClustalW tool (https://www.genome.jp/tools-bin/clustalw). The conserved active site of the enzyme from different species is indicated.
(PDF)

**S2 Fig. Multiple sequence alignment of CK2α orthologous from the indicated species.** Sequences were aligned using the ClustalW tool (https://www.genome.jp/tools-bin/clustalw). The conserved active site of the enzyme from those species is indicated.
(PDF)

**S3 Fig. Multiple sequence alignment of CK2β orthologous from different species as indicated.** Sequences were aligned using the ClustalW tool (https://www.genome.jp/tools-bin/clustalw). *S. pombe* CK2β possess four CK2α target serine residues at the first 14 aminoacid

residues at the N-terminus domain of the polypeptide. Those serine residues are underlined.
(PDF)

**S4 Fig. Multiple sequence alignment of Aurora Kinase (AUK) orthologous from the indicated species.** Sequences were aligned using the Clustal Omega tool (https://www.ebi.ac.uk/Tools/msa/clustalo/). The active site and the activation loop of the enzyme from those species is indicated.
(PDF)

**S5 Fig. Multiple sequence alignment of CK1 protein kinases genes.** All sequences from *T. cruzi* containing shared regions with CK1 were aligned and showed in the figure. The alignment was generated using the NCBI Multiple Sequence Alignment Viewer using the information of the Genbank database. CK1 kinase studied in this work is highlighted in yellow and indicated with a red asterisk. Red: Highly conserved regions. Gray: Middle conserved regions. Blue: Low conserved regions.
(PDF)

**S6 Fig. Unfolded inactive Tcpolβ is not a substrate for CK1α.** Amounts of 50, 100 and 200 ng of inactive polymerase were phosphorylated with 20 pmol of CK1α. It can be observed that unfolded Tcpolβ is not a substrate for the protein kinase. In the bottom of the figure it can be seen a coomassie blue stained SDS-PAGE with the different amounts of the analyzed proteins. (+) indicates control experiment with only 20 pmol of the protein kinase.
(PDF)

**S7 Fig. Unfolded Tcpolβ is not a substrate for TcAUK1.** Amounts of 50 and 100 ng of inactive Tcpolβ were assayed with 20 pmol of TcAUK1. It can be observed that AURKA does not phosphorylate unfolded Tcpolβ. A SDS-PAGE analysis is shown in the bottom of the figure. (-) indicates a negative control without protein kinases. (+) indicates a positive control with only 20 pmol of AURK.
(PDF)

**S8 Fig. *T. cruzi* CK2α does not get auto-phosphorylated.** As indicated on the top of the figure, 15 pmol of *S. pombe* CK2β (pCK2β)(pCK2becated on the tα were used in the assays. It can be observed that TcCK2α does not get auto-phosphorylated, however, it can phosphorylate pCK2β, which has not activity by itself.
(PDF)

**S1 Data. This supporting file shows the complete dataset of all the experiments performed in this work and presented in graphics in all figures.**
(XLSX)

## Author Contributions

**Conceptualization:** Edio Maldonado, Aldo Solari.

**Data curation:** Edio Maldonado, Aldo Solari.

**Formal analysis:** Edio Maldonado, Aldo Solari.

**Funding acquisition:** Aldo Solari.

**Investigation:** Edio Maldonado, Diego A. Rojas, Fabiola Urbina, Aldo Solari.

**Methodology:** Edio Maldonado, Diego A. Rojas, Fabiola Urbina, Aldo Solari.

**Project administration:** Aldo Solari.

**Supervision:** Edio Maldonado, Aldo Solari.

**Validation:** Diego A. Rojas.

**Writing – original draft:** Edio Maldonado, Diego A. Rojas, Aldo Solari.

**Writing – review & editing:** Edio Maldonado, Diego A. Rojas, Fabiola Urbina, Aldo Solari.

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
