## [Decision Letter · Decision Letter 0]

30 Mar 2021

Dear Dr Maldonado,

Thank you very much for submitting your manuscript "T. cruzi DNA polymerase beta (Tcpolβ) is phosphorylated in vitro by CK1, CK2 and AURKA leading to the potentiation of its DNA synthesis activity" for consideration at PLOS Neglected Tropical Diseases. As with all papers reviewed by the journal, your manuscript was reviewed by members of the editorial board and by several independent reviewers. In light of the reviews (below this email), we would like to invite the resubmission of a significantly-revised version that takes into account the reviewers' comments.  

We cannot make any decision about publication until we have seen the revised manuscript and your response to the reviewers' comments. Your revised manuscript is also likely to be sent to reviewers for further evaluation.

Sincerely,

Ziyin Li, Ph.D.

Guest Editor

Margaret Phillips

Deputy Editor

Reviewer's Responses to Questions

**Key Review Criteria Required for Acceptance?**

**Methods**

-Are the objectives of the study clearly articulated with a clear testable hypothesis stated?

-Is the study design appropriate to address the stated objectives?

-Is the population clearly described and appropriate for the hypothesis being tested?

-Is the sample size sufficient to ensure adequate power to address the hypothesis being tested?

-Were correct statistical analysis used to support conclusions?

-Are there concerns about ethical or regulatory requirements being met?

Reviewer #1: The hypothesis proposed by the Authors is that ROS triggers a signaling pathway that activates TbpolB leading to its repair of kDNA in T. cruzi however, the hypothesis is not tested in this manuscript. Rather, the authors performed descriptive biochemical characterization of how the activity of DNA polymerase beta is affected by three T. cruzi kinases CK1, CK2 and AURKA from the parasite T. cruzi. The rationale for this study was that the authors identified consensus phosphorylation sequences for the Ser/Thr kinases casein kinase 1, casein kinase 2 and aurora kinase A in TcpolB from a previous study. Here the authors sought to determine if these kinases were able to phosphorylate TcpolB and the effect this had on its biochemical activity.

Reviewer #2: Metthodology is adequate to answer the questions proposed. 

Minor point:

To reconstitute the CK2 holoenzyme both subunits were mixed at 1:1 molar ratio, renatured by dialysis, loaded on a NTA-Ni-Agarose column and eluted with an Imidazole gradient from 5-250 mM: It would be good to explain how is the renaturation process by dialysis.

**Results**

-Does the analysis presented match the analysis plan?

-Are the results clearly and completely presented?

-Are the figures (Tables, Images) of sufficient quality for clarity?

Reviewer #1: There are several areas where the figures and the presentations need improvement for clarity in the section: “Cloning, expression and functional characterization of T. cruzi CK1”. In presenting Fig. 2 the authors begin discussing Fig. 2A-C and then jump to Fig. 3A and go back to presentation of Fig. 2D/E which is not logical to the reader. In the autoradiograph of Figure 2C, it is clear that casein is a poor substrate for TcCK1a, but it is not clear what the authors declare as activity in Figure 2C as no units are designated in the legend or on the graph. In Fig. 2D there is no explanation of the small bands at the bottom of the gel. 

The section describing: “Cloning, expression and functional characterization of T. cruzi CK2” is confusing to the reader. The reader does not understand why the authors spent so much effort writing about the inability of TcCK2 to phosphorylate casein or TbCK2b instead of directly focusing their attention on its phosphorylation of TbpolB. 

That TBB is and inhibitor of the hybrid or pCK2b is overinterpreted, as it does not show any significant inhibition at any concentration with respect to the control (Fig.3B/C). Likewise, phosphorylation by the hybrid is likely artifact that may lead to misinterpretation or overinterpretation of experimental results in Fig. 3B/C and Fig 3D/E.

In the section: “Effect of pharmacological inhibitors on the activity of TcCK1, TcAURKA and TcCK2”.

The authors do not present the figures in a logical order making it difficult for the reader to follow. The authors start by discussing Fig. 5A-D then they jump back to presentation of Fig.3B and continue with the presentation of Fig. 5E/F.

Reviewer #2: Figure 2D: what is the lower band visualized? 

The molecular mass of CK1 alpha is similar to Tcpolbeta. Then, I’m wondering if phosphorylation signal increasement could be only consequence of auto-phosphorylationhow of CK1 alpha.

Figure S1: red lines, gray box and blue line have to be in the legend.

Figure S4. Although authors say that CK2beta presents active sites and activation loop domain, these regions are not showed in Figure S4.

It might be explained in the text why authors used TBB.

Figure 4: I cant’see discrimination between AURKA and polbeta. It seems to me that the different migration is just consequence of gel run. Again, it is important to discriminate between two bands, especially because AURKA has autophosphorylation activity.

All graphs of quatification says Phosphorilation activity but it is not explained in the figure nor in the legend, which band was quantified.

**Conclusions**

-Are the conclusions supported by the data presented?

-Are the limitations of analysis clearly described?

-Do the authors discuss how these data can be helpful to advance our understanding of the topic under study?

-Is public health relevance addressed?

Reviewer #1: Understanding the mechanisms that regulate the damage/repair of kDNA in Trypanosomes is an interesting and important area of study. In this manuscript, the data is promising, but it is difficult for the reader to come to the same conclusion as the authors given the need for improvement in presentation and experimental design. The authors do not provide a good description about the limitations of the analysis mostly because the presentation of the data and interpretations needs improvements. This work can potentially advance our understanding of the signaling networks involved in regulating or potentiating TbpolB activity, but the presentation and interpretation of the data needs improvement.

Reviewer #2: (No Response)

**Editorial and Data Presentation Modifications?**

Reviewer #1: The manuscript is rife with grammar errors making it difficult to read. There are several free or paid online grammar and editing program that the authors should take advantage of.

Reviewer #2: End of 6th paragraph of introduction: verb is missing: Interestingly, this enzyme exists in two forms, one that IS phosphorylated and other which is not

**Summary and General Comments**

Reviewer #1: In the section: “The activity of TcpolB increases by phosphorylation by TcCK1, TcCK2 and TcAURKA”.

Both counting by scintillation and by autoradiography/SDSPAGE (Fig. 6A/B) are classic assays for qualitatively measuring polymerase activity. However, just increasing the amount of enzyme (kinase) in an assay is not sufficient proof to demonstrate that the activity of the TcpolB increases with phosphorylation. A better conclusion could be made by a quantitative enzyme kinetic analysis of TcpolB to determine if phosphorylation by any of the TcKinases tested increases its specific activity or turnover.

Reviewer #2: Authors investigated the signal trasnduction involved with POlbeta activity, which is a relevant topic important for our understand about cell DNA repair in T. cruzi. The involvment of specific enzymes in this process is very interesting. However, as I said in the result section, I'm affraid that some results might be carefully interpreted.

PLOS authors have the option to publish the peer review history of their article (what does this mean?). If published, this will include your full peer review and any attached files.

Reviewer #1: No

Reviewer #2: No
---

## [Decision Letter · Decision Letter 1]

28 May 2021

Dear Dr Maldonado,

Thank you very much for submitting your manuscript "T. cruzi DNA polymerase beta (Tcpolβ) is phosphorylated in vitro by CK1, CK2 and AURKA leading to the potentiation of its DNA synthesis activity" for consideration at PLOS Neglected Tropical Diseases. As with all papers reviewed by the journal, your manuscript was reviewed by members of the editorial board and by several independent reviewers. The reviewers appreciated the attention to an important topic. Based on the reviews, we are likely to accept this manuscript for publication, providing that you modify the manuscript according to the review recommendations. 

Sincerely,

Ziyin Li, Ph.D.

Guest Editor

Margaret Phillips

Deputy Editor

Reviewer's Responses to Questions

**Key Review Criteria Required for Acceptance?**

**Methods**

-Are the objectives of the study clearly articulated with a clear testable hypothesis stated?

-Is the study design appropriate to address the stated objectives?

-Is the population clearly described and appropriate for the hypothesis being tested?

-Is the sample size sufficient to ensure adequate power to address the hypothesis being tested?

-Were correct statistical analysis used to support conclusions?

-Are there concerns about ethical or regulatory requirements being met?

Reviewer #1: The work in this manuscript does not pursue a clearly stated testable hypothesis. The authors identified consensus phosphorylation sequences for the Ser/Thr kinases casein kinase 1, casein kinase 2 and aurora kinase A in TcpolB. Here the authors sought to determine if these kinases were able to phosphorylate TcpolB and the effect this had on its biochemical activity.

Reviewer #2: All questions were properly answered

**Results**

-Does the analysis presented match the analysis plan?

-Are the results clearly and completely presented?

-Are the figures (Tables, Images) of sufficient quality for clarity?

Reviewer #1: The authors Figure 2 the authors show that TcpolB is a substrate for TcCK1a and that heparin can inhibit this activity.

In Figure 3 the authors show that TcCK2b is a activity simulator for TcCK2a phosphorylation of TcpolB. It is difficult to understand the relevance of the hybrid-holoenzyme studies other than to show the conservation of the TcCK2b subunit in the holoenzyme. Also Fig.3D is not mentioned in the same part of the results section as the rest of Fig. 3 A-F.

In Figure 4, the authors show that TcAURKa can phosphorylate TcpolB. However assuming that the AUR concentration remains constant, the appearance of two bands is not consistent with one being TcpolB and one being TcAURKA. Could the doublet represent two species of TcpolB phosphorylation? Fig. 1 shows that TcpolB has two AURKA consensus sites.

Fig.6 is a qualitative assay that shows TcpolB gets activated by three kinases.

Reviewer #2: yes

**Conclusions**

-Are the conclusions supported by the data presented?

-Are the limitations of analysis clearly described?

-Do the authors discuss how these data can be helpful to advance our understanding of the topic under study?

-Is public health relevance addressed?

Reviewer #1: The authors show that TcpolB is phosphorylated by three kinases and that its activity is increased by phosphorylation in vitro. Exploring which of kinases are in the signalling cascade that regulates TcpolB activity is an interesting and important area of study but the data in this study is still too preliminary to conclude that TcpolB is part of the signaling cascade for one of the kinases. This work can potentially advance our understanding of the signaling networks involved in regulating or potentiating TcpolB activity, but the authors need to do a better job of developing a hypothesis for which of these kinases regulates TcpolB in T. cruzi.

Reviewer #2: yes

**Editorial and Data Presentation Modifications?**

Reviewer #1: Fig 5 is not logically organized, it would be less confusing to the reader if the authors put all of the inhibitor information from the manuscript in this figure or include inhibitor information in each relevant figure. This would avoid the reader bouncing back to result sections of the manuscript that have already been covered.

Reviewer #2: (No Response)

**Summary and General Comments**

Reviewer #1: (No Response)

Reviewer #2: (No Response)

PLOS authors have the option to publish the peer review history of their article (what does this mean?). If published, this will include your full peer review and any attached files.

Reviewer #1: No

Reviewer #2: No

Figure Files:

Data Requirements:

Reproducibility:

References

---

## [Editor Report · Decision Letter 2]

15 Jun 2021

Dear Dr Maldonado,

Thank you very much for submitting your manuscript "T. cruzi DNA polymerase beta (Tcpolβ) is phosphorylated in vitro by CK1, CK2 and AURKA leading to the potentiation of its DNA synthesis activity" for consideration at PLOS Neglected Tropical Diseases. 

The handling editor has identified an error in the revised manuscript that needs to be corrected before the manuscript can be accepted. The manuscript described a homolog of Aurora A kinase in T. cruzi, however, after careful examination of the sequence and the database, the handling editor found that the designated Aurora A kinase (AURKA) is actually the Aurora B kinase, which was named AUK1 and has a close homolog in Trypanosoma brucei (TbAUK1). Therefore, it is strongly encouraged that the authors correct this nomenclature to avoid confusion. There is no Aurora A kinase homolog in kinetoplastid parasites, as there is no spindle pole body structure that requires an Aurora A kinase function. Please check the manuscript text and figures carefully to make changes.

Sincerely,

Ziyin Li, Ph.D.

Guest Editor

Margaret Phillips

Deputy Editor

Figure Files:

Data Requirements:

Reproducibility:

References

---

## [Editor Report · Decision Letter 3]

23 Jun 2021

Dear Dr Maldonado,

We are pleased to inform you that your manuscript 'T. cruzi DNA polymerase beta (Tcpolβ) is phosphorylated in vitro by CK1, CK2 and TcAUK1 leading to the potentiation of its DNA synthesis activity' has been provisionally accepted for publication in PLOS Neglected Tropical Diseases.

Best regards,

Ziyin Li, Ph.D.

Guest Editor

Margaret Phillips

Deputy Editor

---

## [Editor Report · Acceptance letter]

9 Jul 2021

Dear Dr Maldonado,

We are delighted to inform you that your manuscript, "T. cruzi DNA polymerase beta (Tcpolβ) is phosphorylated in vitro by CK1, CK2 and TcAUK1 leading to the potentiation of its DNA synthesis activity," has been formally accepted for publication in PLOS Neglected Tropical Diseases.

Best regards,

Shaden Kamhawi

co-Editor-in-Chief

Paul Brindley

co-Editor-in-Chief
